# Improving Riparin-A Dissolution through a Laponite Based Nanohybrid

**DOI:** 10.3390/pharmaceutics15082136

**Published:** 2023-08-14

**Authors:** Duanne Mendes Gomes, Lyghia Maria Araújo Meirelles, Paulo Monteiro Araujo, Rayran Walter Ramos de Sousa, Paulo Michel Pinheiro Ferreira, Stanley Juan Chavez Gutierrez, Maria das Graças Freire de Medeiros, Fernanda Nervo Raffin

**Affiliations:** 1Post Program on Pharmaceutical Sciences, Federal University of Piauí—UFPI, Teresina 64049-550, Piauí, Brazil; duannemendesg@gmail.com (D.M.G.);; 2Department of Pharmacy, Federal University of Piauí—UFPI, Teresina 64049-550, Piauí, Brazil; 3Health and Quality of Life Research Laboratory (LAPESQV), University Center Santo Agostinho—UNIFSA, Teresina 64049-550, Piauí, Brazil; 4Laboratory of Experimental Cancerology (LabCancer), Department of Biophysics and Physiology, Federal University of Piauí—UFPI, Teresina 64049-550, Piauí, Brazilpmpf@ufpi.edu.br (P.M.P.F.); 5Post—Program on Development and Technological Innovation in Medications, Federal University of Rio Grande do Norte—UFRN, Natal 59012-570, Rio Grande do Norte, Brazil

**Keywords:** Riparin-A, Laponite, nanohybrid, dissolution, cytotoxicity

## Abstract

(1) Background: Riparin-A presents several pharmacological activities already elucidated, such as antimicrobial modulator, antileishmania, anxiolytic, anti-inflammatory, antinociceptive, and antioxidant. Even with important bioactive effects, the applicability of Riparin-A is limited due to its low solubility in water, impairing its dissolution in biological fluids. Thus, the objective of this study was to develop a nanohybrid based on Riparin-A and Laponite to obtain a better dissolution profile and evaluate its cytotoxic potential. (2) Methods: The formation of a hybrid system was highlighted by X-ray powder diffraction, infrared spectroscopy, and thermal analysis. Solubility, dissolution, and cytotoxicity studies were performed; (3) Results: An increase in the solubility and aqueous dissolution rate of Riparin-A was observed in the presence of clay. Diffractometric analysis of the hybrid system suggests the amorphization of Riparin-A, and thermal analyses indicated attenuation of decomposition and melting of the Riparin-A after interaction with clay. Furthermore, the nanosystem did not exhibit cytotoxic activity on normal and tumorigenic lines. (4) Conclusions: These results are promising for the development of the Riparin-A/Laponite nanosystem for therapeutic purposes, suggesting an increase in the range of possible routes of administration and bioavailability of this bioactive compound.

## 1. Introduction

The use of natural substances as therapeutic agents has evolved with the isolation of chemicals and the possibility of their biosynthesis [1]. Modifications of molecules derived from medicinal plants to obtain new structures with pharmacological activity and less adverse effects are the focus of several studies that intend to enable their therapeutic application. Among these are the synthetic prototypes isolated from the green fruit of *Aniba riparia* (Nees) Mez, Lauraceae, a typical plant from the Amazon region, where it is popularly known as “louro” [2,3]. *A. riparia* is of great interest as a medicinal plant due to its chemical composition of alkaloids of the alkamide type known as riparins, which have an amide function formed from the condensation of a saturated fatty acid and an amine group, with few representatives in nature [4].

Since the synthetic route has become the main way of obtaining molecules from this group of compounds, several structural derivatives have been produced, including Riparins A and B. Additionally, from the condensation of methyl esters with substituted phenylethylamines, synthesized Riparins C, D, E, and F can be obtained using the Schotten–Baumann reaction, a simple and reliable method that enables the commercial exploitation of these molecules by the pharmaceutical industry [3].

Among the isolated substances, Riparin-A (Rip-A) stands out for its potential for several pharmacological activities, such as antimicrobial modulator against strains that overexpress the NorA efflux pump [5], anti-inflammatory [6], antinociceptive [6], anxiolytic [7], antioxidant [3,8,9], leishmanicidal activity on promastigotes forms of *Leishmania amazonensis* and antitumor potential in colon adenocarcinoma cells [8,10].

Even with important bioactive effects, the applicability of Rip-A is limited due to its low solubility in water, impairing its dissolution in biological fluids. In this context, the use of technologies aimed at improving physical-chemical properties stands out. Several strategies have been proposed, with the aim of increasing the solubility and consequent enhancement in the dissolution rate of poorly soluble drugs: inclusion complexes with cyclodextrins [11], nanostructured systems with mineral derivatives of clays [12], solid dispersions based on hydrophilic polymers [13], particle size reduction and use of carriers [14].

Laponite is a synthetic nanoclay, more specifically, synthetic lamellar silicate, formed from inorganic minerals; it belongs to the smectite class and has the appearance of a fine white powder, with great potential in the development of drug delivery systems due to its defined chemical composition, free of impurities (common in natural clays), biocompatibility, and hydrophilicity, among other properties [15,16]. The interaction between the nanoclay and active substances in the systems can occur in several ways, such as the emission of the active substance being carried in interparticle sites, interlayer sites or can be adsorbed on the surface or on the edges, depending on the size and charge of the drug substances [17].

It consists of stacks of aggregated nanodisks, interacting strongly with various types of chemical entities, from small molecules or ions, natural or synthetic polymers, and different inorganic nanoparticles [18]. Based on these interactions, clay minerals and their modified forms can be effectively used to modify drug release based on their adsorption capacity, swelling, and colloidal properties [19,20].

Hybrid systems based on clays have attracted the interest of studies due to the advantages provided regarding physicochemical properties, such as controlled release, masking of unpleasant organoleptic characteristics of drugs, better stability, the possibility of a system with thermoresponsive properties, or direct drug release to a target site and minimize its toxic effects [15,16,21]. In addition, water solubility can be increased by forming hybrid systems with clay derivatives [12,22].

In view of this, the expectation is to combine the benefits of nanotechnology with promising hybrid systems capable of circumventing pharmacokinetic and pharmacodynamic limitations presented in conventional formulations [23]. In the literature, there are some examples of drug–clay hybrids based on laponite, such as nanohybrids using itraconazole, metformin, and griseofulvin, in addition to the development of drug–clay–polymer [21,24,25].

In short, the objective of the study was to prepare a nanohybrid based on Riparin-A and Laponite (Rip-A/Lap) in order to obtain a better dissolution profile, perform its physical-chemical characterization, drug release profile, and evaluate its cytotoxic potential.

## 2. Materials and Methods

### 2.1. Materials and Reagents

Rip-A was synthesized following the Schoten-Bauman reaction according to the method described in [26] and kindly provided by Prof. Dr. Stanley Juan Chavez Gutierrez (Department of Pharmacy, UFPI). Laponite XLG samples were provided by BYK Additives & Instruments. The other reagents used were of analytical grade.

### 2.2. Factorial Design for Preliminary Investigation of Riparin-A Solubility

A factorial design was used as a rational method to evaluate the effect of each factor (Laponite concentration and agitation time) on the water solubility of Rip-A in excess, analyzing the Laponite concentrations (0, 0.4, and 0.8%) and stirring time (24, 48, and 72 h) followed by syringe filter (0.22 µm). The time was defined based on other authors that prepared composites between laponite and itraconazole [12]. In this way, nine different experiments were prepared, the factors being studied at three levels (Table 1).

Analysis of variance (ANOVA) was used to verify the effects of the selected factors on the response of the experiment. Statistical analysis of the factorial design was performed using the STATISTICA version 6.0 software.

### 2.3. Solubility Studies

To evaluate the effect of clay on the solubility of the active ingredient, Rip-A was added in excess in microtubes containing 1 mL of aqueous dispersions with increasing concentrations of laponite (0.0–1.4%, *w*/*v*). The samples were kept under constant agitation for 48 h at room temperature, using a tube homogenizer (model AP-22, Phoenix-Luferco) [17].

After the stirring time, the samples were centrifuged (Daiki, Tokyo, Japan, DT-5000) under 9000 rpm for 10 min to separate the supernatant, followed by filtration on 0.22 µm membranes and analyzed in a UV-Vis spectrophotometer (Shimadzu, Kyoto, Japan, UV—1800) to quantification of Rip-A at a wavelength of 233 nm, using distilled water as a blank [9]. The experiments were performed in triplicate for each Laponite concentration tested.

### 2.4. Preparation of the Physical Mixture and Nanohybrid

The physical mixture (MF) was obtained by homogenizing 1 g Rip-A and 10 g Lap for 30 min in a porcelain mortar, with the aid of a pestle, in the proportion of 1:10 (*w*/*w*), defined in the solubility study and stored in a desiccator.

With the proportions defined in the solubility study, the ethanolic solution of Rip-A and aqueous dispersion of Lap (1:10) were mixed at room temperature. The samples were homogenized with a magnetic stirrer for 48 h and lyophilized (Solab, Paris, France, SL-152) under previously adjusted conditions. The composites were stored in a desiccator.

### 2.5. Solid State Characterization of Physical Mixture, Nanohybrid and Raw Materials

The Lap, Rip-A, MF, and Rip-A/Lap nanohybrid samples were characterized using thermal analysis, X-ray diffraction, and Fourier-transform infrared spectroscopy.

#### 2.5.1. Thermogravimetric Analysis (TGA) and Differential Scanning Calorimetry (DSC)

The thermogravimetric analysis was carried out in an analyzer (TGA-50, Shimadzu, Kyoto, Japan), using 5.0 ± 0.5 mg of sample contained in an aluminum crucible, heated at a rate of 10 °C/min, in an atmosphere of nitrogen, with a gas flow of 100 mL/min, in a temperature range of 50–400 °C.

Differential scanning calorimetry was performed using 3.0 ± 0.5 mg of sample in closed crucibles, heated between 50–400 °C, at 10 °C/min, under nitrogen atmosphere to obtain the DSC curves (DSC-2920, TA Instruments, New Castle, DE, USA).

#### 2.5.2. X-ray Powder Diffraction (XRPD)

The diffractograms of the samples were obtained using an X-ray diffractometer (Rigaku, Tokyo, Japan, Miniflex model) (λ = 1.5418 Å). The diffractograms were obtained with a 2 θ angle ranging from 5–50° at a rate of 2°/min.

#### 2.5.3. Fourier-Transform Infrared (FTIR) Spectroscopy

The spectra of the samples were obtained in the mid-IR region (4000 to 650 cm^−1^) in a spectrometer (Cary 630, Agilent) using the accessory for attenuated total reflectance (RTA). About 5 mg of each sample was placed directly on the surface of the RTA accessory, applying light pressure to promote greater contact between the solid sample and the crystal surface. The spectra were obtained from 64 scans, with a resolution of 4 cm^−1^.

### 2.6. In Vitro Dissolution Study

A dissolution study was carried out with the paddle apparatus (Mod. 299, Ethik Technology, Vargem Grande Paulista, Brazil) at 50 rpm and 37.0 ± 0.5 °C. A mass equivalent to 5 mg of Rip-A and the nanohybrid were added to 900 mL of distilled water as a dissolution medium. Aliquots of 5 mL of the medium were collected at 5, 10, 15, 20, 30, 45 and 60 min. Samples were filtered and analyzed using UV-VIS spectrophotometry (Shimadzu UV—1800) at 233 nm. Dissolution was determined based on dissolution efficiency over 60 min. The experiments were performed in triplicate.

### 2.7. In Vitro Cytotoxicity on Tumor and Normal Cells

The murine cell lines B-16/F-10 (melanoma) and L-929 (fibroblast) were provided by the National Cancer Institute (Bethesda, MD, USA). They were incubated in Roswell Park Memorial Institute medium (RPMI 1640) and Dulbecco’s Modified Eagle Medium (DMEM), respectively, supplemented with 10% fetal bovine serum, 1% antibiotics (streptomycin/penicillin, Sigma^®^, St. Louis, MO, USA), and 5 mg/L of amphotericin (Cristalia^®^, Fair Oak, UK) and maintained in 5% CO_2_ at 37 °C (Shel Lab^®^, Cornelius, OR, USA). Rip-A, Lap, and Rip-A/Lap samples were diluted in dimethyl sulfoxide (DMSO) P.A (Dinâmica^®^, London, UK), and a 10 mg/mL stock solution of each sample was prepared. Thus, lines were plated at 5 × 10^3^ cells/m and treated with the substances (1.95–250 µg/mL for hybrid and Lap and 0.195–25 µg/mL for Rip-A). The plates were incubated for 72 h at 5% CO_2_ at 37 °C. At the end of this period, 20 µL of tetrazolium salt solution (MTT) solubilized in sterile phosphate saline buffer (PBS) was added, and the plates were incubated for an additional 4 h. Formazan crystals were dissolved in non-sterile DMSO and the absorbance was read at 560 nm (GloMax^®^ Explorer Multimode Microplate Reader).

### 2.8. Statistical Analysis

Differences were evaluated by comparing data using one-way analysis of variance (ANOVA) followed by the Newman–Keuls test (*p* < 0.05) using the GraphPad software, version 9.0. All in vitro studies were carried out in duplicate and represented independent biological evaluations.

## 3. Results and Discussion

### 3.1. Factorial Design for Preliminary Investigation of Riparin—A Solubility

Figure 1 shows Rip-A UV-Vis absorption spectra in increasing concentrations, from 0.2 up to 1.0 mM. Rip-A presented maximum absorption at 233 nm in hydroalcoholic solution (1:9), increasing absorbance as the concentration increased. From the calibration curve of Rip-A, it was obtained the equation y = 1.1062x + 0.0229, with a correlation coefficient of 0.9996, which was used to calculate Rip-A concentration in subsequent quantifications. The method was linear, with the angular coefficient statistically different from zero.

The effects of the parameters varied to obtain the Rip-A/Lap nanohybrid in the factorial design experiments are depicted in the Pareto diagram (Figure 2). The results showed that the increase in Rip-A aqueous solubility was influenced by the concentration of laponite. The Pareto chart indicates that, for the variables analyzed, only the variation in the clay concentration was statistically significant in the improvement in Rip-A solubility. There was no statistically significant difference between the analyzed time intervals (24 h, 48 h, and 72 h), showing that adsorption equilibrium occurs quickly.

The influence of keeping Rip-A under agitation in increasing concentrations of Lap can be seen in Figure 3; it was obtained an increase in the aqueous solubility of Rip-A, suggesting a possible interaction with the clay. There was a rise of 231% in the aqueous solubility of Rip-A added to the 1.2% Lap dispersion after 48 h of agitation, considering the intrinsic solubility of Rip-A 1.78 mM. According to Jung et al., the aqueous solubility of drug can be increased when it is intercalated in a hydrophilic clay mineral [12]. The use of concentrations greater than 1.2% Lap was not possible due to the change in the texture of the dispersion, which assumed a gel-like appearance, making it difficult to homogenize the system.

Lap presents a lamellar, disc-like shape (ca. 25 nm in diameter and 1 nm in thickness) with negative charges distributed on the faces (OH^−^) and positive charges on the edges (Na^+^) that react with hydroxide ions in water. After ion dissolution, the nanoclay particles interact with each other, and sodium ions diffuse to the surfaces of the galleries formed, resulting in an expanded thixotropic gel with a “house-of-cards” structure that holds the drug [27]. Moreover, Lap interacts with charged, polar, and nonpolar species due to its large charged surface, which enables the adsorption of ions or molecules via ion exchange, van der Waals forces, hydrogen bonding, and cation/water bridges, as well as protonation and ligand exchange at the edges of the crystal [19]. NH and CO groups in the Rip-A structure provide some polarity to the molecule despite the low solubility in water. The interaction between Rip-A and Lap could occur via different mechanisms, which will be suggested from the analysis of the FTIR spectrum.

Similar solubility values were reported using β-cyclodextrin (β-CD) as a carrier organic macromolecular. Promising results were achieved via the Rip-A/β-CD inclusion complex, with an increase in the aqueous solubility of Rip-A of 240.2% in relation to the intrinsic solubility [11]. These results were obtained for samples prepared using spray drying, a scalable method for the industry [28]. However, the need for a large amount of cyclodextrin, due to its high molecular weight, is a limitation to its use in formulations involving high drug doses.

### 3.2. Solid State Characterization of Physical Mixture, Nanohybrid and Raw Materials

#### 3.2.1. Thermogravimetric Analysis (TGA) and Differential Scanning Calorimetry (DSC)

Thermoanalytical techniques allow for verifying changes resulting from thermal events related to degradation and alteration of stability [29]. The thermogravimetric profiles of the samples are represented in Figure 4. For comparison purposes, a physical mixture (MF) containing Rip-A and Lap was also analyzed.

As expected, the clay shows high thermal stability, with an initial mass loss attributed to the release of water molecules absorbed by the samples [30]. Lap mass loss occurred mainly up to 120 °C. This event is related to the removal of water from the nanosilicate structure [31,32,33]. Lap had a mass loss of approximately 15% at the event, but the mass loss by dehydration of the hybrid was lower than that observed for the physical mixture (MF), probably due to the replacement of water molecules by Rip-A in the clay structure during the formation of the hybrid. MF and nanohybrid had a loss of 12% and 7% up to 400 °C, respectively.

DSC curve of Rip-A (Figure 5) showed a characteristic, intense, and well-defined endothermic peak at 116.7 °C, which is attributed to the melting of the substance in its crystalline form, a result consistent with that reported by Araújo et al. [11]. In the Lap DSC curve, a characteristic peak of water loss was observed at around 100 °C. The intensity of the melting peak is reduced in the MF despite being slightly superimposed on the peak corresponding to the dehydration of Lap. A marked decrease in the Rip-A/Lap hybrid curve indicates the possible formation of the nanosystem by the confinement of the drug between the clay sheets.

#### 3.2.2. X-ray Powder Diffraction (XRPD)

The diffractograms of the samples are represented in Figure 6 and show the crystallinity of Rip-A (Figure 6a). The diffraction profile of the Rip-A sample presents an intense reflection at angle 2θ = 12.6°, followed by other smaller reflections at 6.8°, 18.9°, 21.78°, 38.46°, and 45.3°. Lap shows intense peaks at 6.06°, 19.7°, and 35.4° (Figure 6b). Rip-A/Lap hybrid showed more intense peaks at 6.38°, 17.3°, and 28.2°. This displacement and intensity difference in relation to Lap peaks can be associated with the interaction with the Rip-A molecule. The absence of the characteristic peaks of Rip-A in the nanohybrid suggests intercalation of the biomolecule, with changes to an amorphous pattern. MF showed a diffraction profile very similar to Lap, with peaks of Rip-A suppressed by Laponite, which is in greater proportion.

#### 3.2.3. Fourier-Transform Infrared (FTIR) Spectroscopy

The chemical structure of Rip-A was confirmed using characteristic bands in the FTIR spectra at 3350 cm^−1^ (stretching N–H), 2900 cm^−1^ (stretching C–H sp^2^), 1640 cm^−1^ (stretching C=O), and in the range of 1600 to 1450 cm^−1^ (stretching C=C) (Figure 7). Similar Rip-A spectra were obtained by Araújo et al. [11]. On the other hand, the vibration bands characteristic of the clay were assigned at 3370 cm^−1^ (stretching O–H), 1640 cm^−1^ (torsion O–H), and 960 cm^−1^ (stretching de Si–O). Such data corroborate the Lap spectrum obtained by Zhou et al. [34].

It is possible to identify the characteristic bands of Lap both in the MF and in the nanohybrid spectra. However, the bands assigned to the active ingredient were suppressed due to the low proportion of Rip-A used in the preparation of the hybrid.

In addition, the absence of the O–H stretching band in the Rip-A/Lap nanohybrid spectrum in comparison to the Lap one indicates a considerable loss of hydrogen bonds; however, there is also an absence of the free O–H stretching band, suggesting interactions between the functional groups of clay and drug. A decrease in peak intensity was also observed in the Si–O stretching band in the spectrum of Rip-A/Lap compared to Lap, showing the functional groups of the clay involved in the interaction with Rip-A.

### 3.3. In Vitro Dissolution Study

The dissolution curves of the free Rip-A and Rip-A/Lap systems are shown in Figure 8. The nanosystem presented a superior aqueous dissolution profile at all analyzed times, presenting a greater rate and more efficient release when compared to free Rip-A. Statistical analysis using *t* test paired with time showed a significant difference between the curves, with a *p* value of 0.0003. At the end of the experiment, the samples showed 61.1% (free Rip-A) and 72.9% (Rip-A/Lap nanohybrid) of the loaded riparian percentage in solution, demonstrating a superior drug release from the nanosystem. Moreover, Rip-A reached the maximum amount in solution after 60 min of the experiment, whereas the hybrid system hit a plateau in about 30 min.

The dissolution profile of Rip-A/Lap nanohybrid compared to free Rip-A corroborates the formation of a more soluble system. The improvement in the dissolution rate is due to the interaction between Rip-A and Lap in the nanohybrid system and can be attributed to the increased solubility conferred by the presence of the clay, in addition to the secondary effects, increased surface area, or wettability [12]. The result is promising, as the dissolution rate can be a limiting factor in the absorption of a bioactive substance, especially if it has low aqueous solubility [35]. Furthermore, it is believed that the drug release profile obtained in the dissolution study may be related to the gradual desorption of Rip-A from the Lap disks.

Comparing the dissolution efficiency (DE%) obtained by calculating the area under the dissolution curves using the trapezoid method [36], it is evident that beyond occurring faster, the dissolution from the nanohybrid presented a DE% of about 1.5 times greater than free Rip-A. This result is very interesting as a strategy to overcome problems in drug bioavailability classified in the class II group in the Biopharmaceutical Classification System [35], where the dissolution step is critical due to the low solubility and slow dissolution of these drugs.

Ferreira et al. showed 68.9% of dissolved Rip-A from the inclusion complex with beta-cyclodextrin. The system developed by our group presents results similar to those achieved by Ferreira et al. However, the conversion of the system into a product can be hampered by the high molecular weight of beta-cyclodextrin, limiting the production of solid pharmaceutical forms, such as capsules and tablets.

Several studies point to synthetic clays as a strategy to improve drug solubility in biological fluids; those systems can be based on drug–clay or drug–polymer–clay. Takahashi and Yamagushi evaluated the effect of Lap on the solubility of griseofulvin, a neutral and water-insoluble molecule [25]. A notable increment in the solubility of the antifungal drug was obtained in the presence of increasing concentrations of laponite, which was attributed to the adsorption of drug molecules to the surface of the solid clay under the amorphous state.

Câmara and Cols developed nanosystems based on PEO-PPO-PEO copolymers, Lap, and β-lapachone (BLPC). BLPC is very poorly soluble in water, which limits its administration, bioavailability, and clinical applications in vivo. In order to overcome this limitation, a hybrid system between drug–polymer–clay was designed. The hybrid system presented an increase of more than 50 times in the aqueous solubility of BLPC [37].

The design of hybrids based on Lap and lipophilic drugs has advantages such as better bioavailability, controlled release, biocompatibility, higher stability, optimizing pharmacological properties, directing drug release to the target site, and minimizing its toxic effects [12,15,21,22,25].

### 3.4. In Vitro Cytotoxicity on Tumor and Normal Cells

*In vitro* cytotoxicity studies are widely used as they allow a preliminary investigation of the cellular toxicity of several compounds [38]. The MTT method can determine the viability and the metabolic state of the cell, allowing it to easily define the cytotoxicity but not the mechanism of action involved [39].

Rip-A/Lap, Rip-A, and Lap showed weak cytotoxic potential on both murine melanoma and fibroblast cells (Figure 9). Only higher concentrations of Rip-A alone (25 µg/mL) and 125 and 250 µg/mL for Lap and Rip-A/Lap displayed antiproliferative effects (*p* < 0.05), but the nanohybrid system based on Rip-A and Lap did not increase the cytotoxic action of Rip-A (*p* > 0.05).

Another study that evaluated the cytotoxicity of Lap corroborates these results. It was observed that the clay did not show antiproliferative potential for human lung fibroblast cells [40] and even increased cell viability when compared to the substances tested alone [41].

Additionally, a hybrid containing Lap also did not show cytotoxicity in a liver cancer cell line (Hep-G2), which reinforces the biocompatibility of laponite and derived products and emphasizes the possibility of different biomedical applications [18,42]. In this sense, the cellular cytotoxicity assays provide subsidies for in vivo toxicity [43].

## 4. Conclusions

A factorial design was applied to optimize the parameters for obtaining a nanohybrid system based on Rip-A and Lap and detected that the increase in clay concentration was directly related to the increase in Rip-A solubility up to a certain amount, above which there was gelation of the sample.

The formation of the nanohybrid system based on Rip-A and Lap was confirmed using the reduced thermal decomposition, amorphization observed on XRD, and changes in the FTIR spectra. Furthermore, the solubility study confirmed an increase in water solubility of Rip-A in the presence of clay. The nanohybrid showed a higher and more efficient aqueous dissolution rate than that presented using Rip-A alone.

Biocompatibility was maintained for normal and tumorigenic cell lines (murine melanoma cells). The characteristic of non-toxicity, together with the increased water solubility of Rip-A, provides perspectives to new research based on the use of Rip-A as an active ingredient for pharmaceutical purposes by the possibility of solving one of the biggest obstacles of this molecule, its low solubility in aqueous medium.

## Figures and Tables

**Figure 1 pharmaceutics-15-02136-f001:**
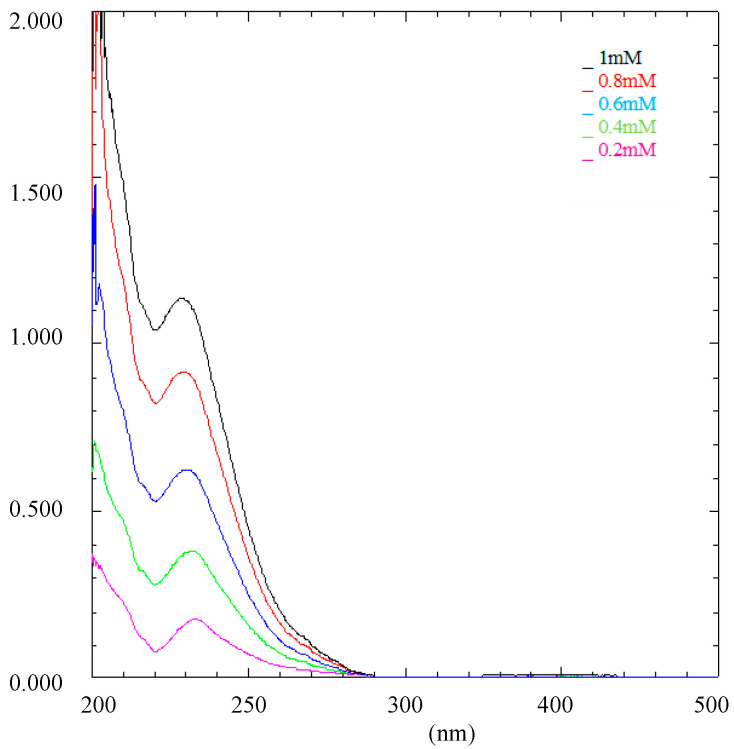
UV-Vis spectrophotometric scanning of Rip-A at different concentrations.

**Figure 2 pharmaceutics-15-02136-f002:**
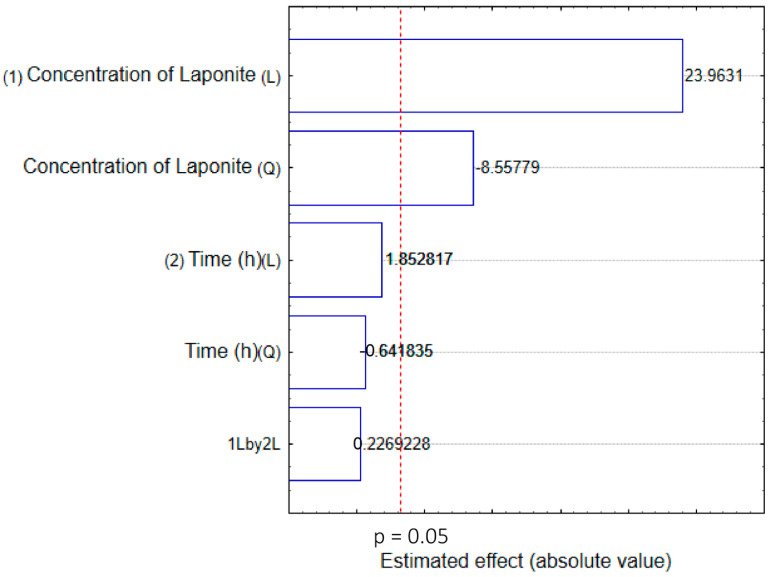
Pareto diagram for the effect of the parameters Laponite concentration and stirring time during nanohybrid (Rip-A/Lap) preparation on the Rip-A solubility.

**Figure 3 pharmaceutics-15-02136-f003:**
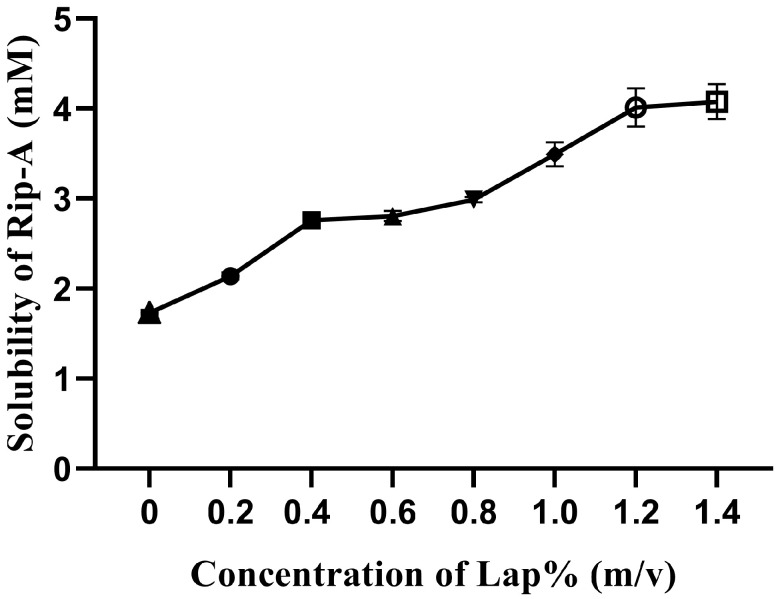
Solubility of Rip-A in the presence of increasing Lap concentrations (0–1.4%, *w*/*v*).

**Figure 4 pharmaceutics-15-02136-f004:**
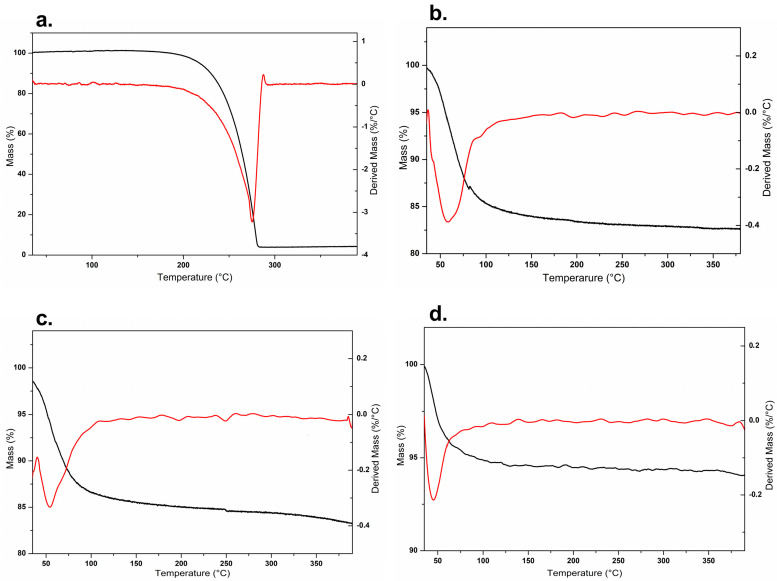
TGA/DTG profiles of Rip-A (**a**), Lap (**b**), MF (**c**), and Rip-A/Lap (**d**).

**Figure 5 pharmaceutics-15-02136-f005:**
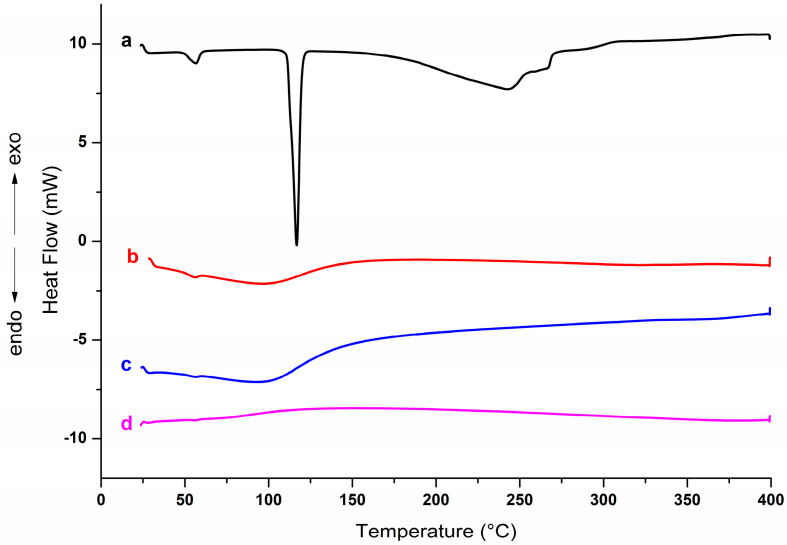
DSC curve of Rip-A (a), Lap (b), MF (c), and Rip-A/Lap (d).

**Figure 6 pharmaceutics-15-02136-f006:**
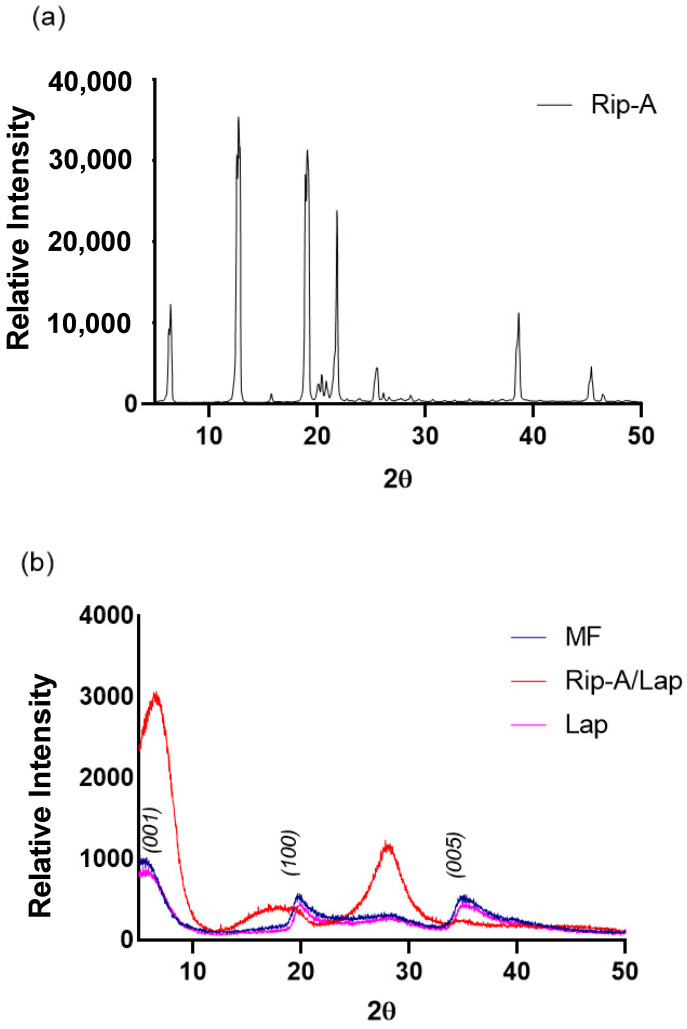
X-ray powder diffraction pattern of the Rip-A (**a**), and Lap, MF, and Rip-A/Lap (**b**).

**Figure 7 pharmaceutics-15-02136-f007:**
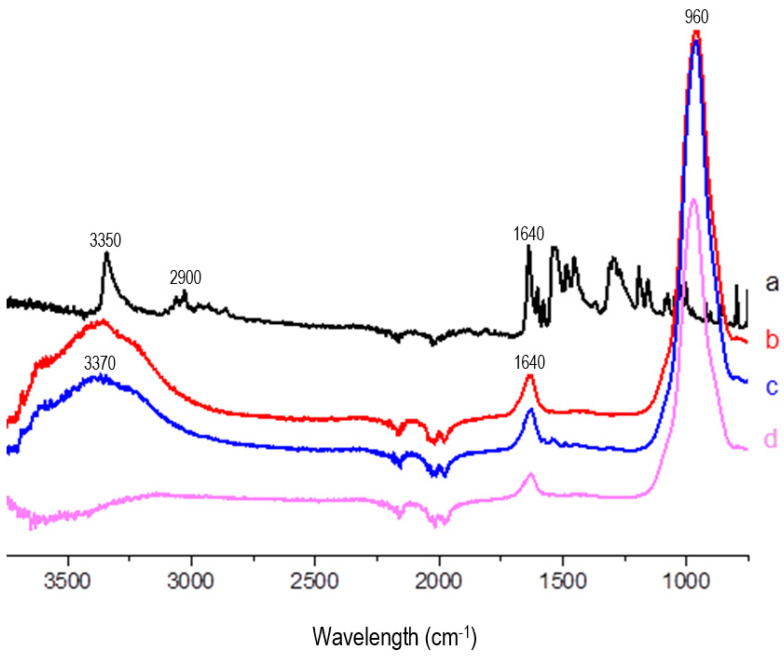
FTIR spectra of Rip-A (a), Lap (b), MF (c), and nanohybrid Rip-A/Lap (d).

**Figure 8 pharmaceutics-15-02136-f008:**
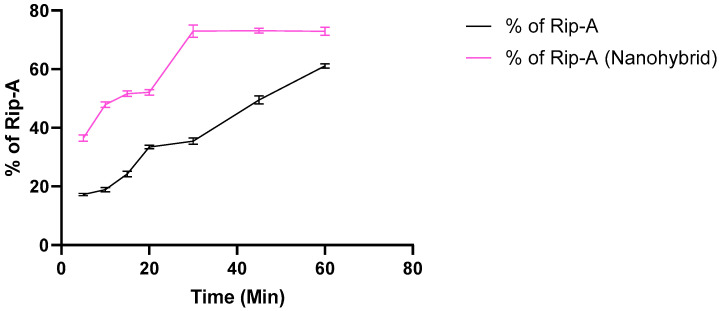
Dissolution curve of free Rip-A and nanohybrid Rip-A/Lap.

**Figure 9 pharmaceutics-15-02136-f009:**
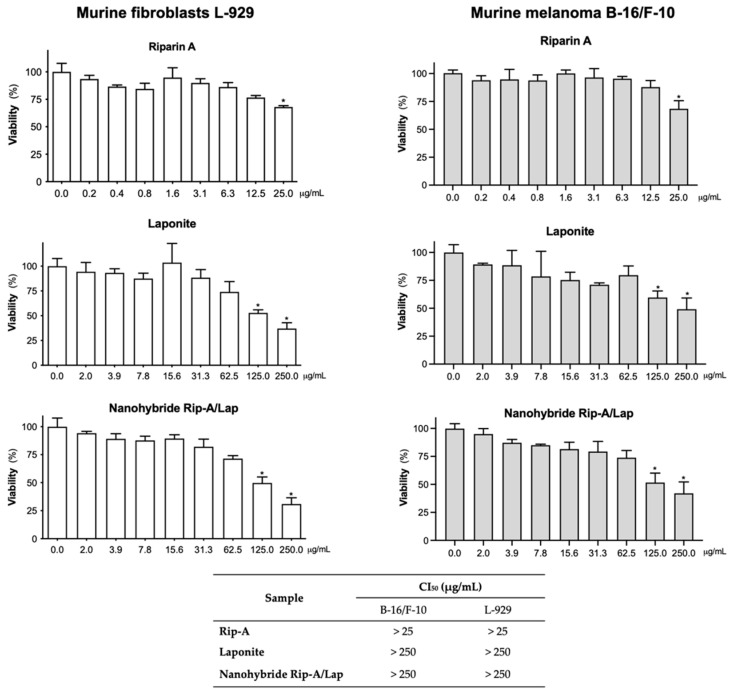
In vitro cytotoxic activity of Rip-A, laponite, and nanohybrid on normal and tumor cell lines determined using the MTT assay after 72 h of incubation. * *p* < 0.05 compared to the vehicle by ANOVA, followed by the Newman–Keuls test.

**Table 1 pharmaceutics-15-02136-t001:** Factors and levels applied to factorial design.

Factors	Levels
A: Laponite concentration (%)	0.0 0.4 0.8
B: Stirring time (hour)	24 48 72

## Data Availability

Not applicable.

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
