# Peer review of "Improving Riparin-A Dissolution through a Laponite Based Nanohybrid"

_pharmaceutics, 2023, doi:10.3390/pharmaceutics15082136_

Round 1

Reviewer 1 Report

The results are novel however the discussion and the explanation of the results are weak. Also, to my opinion cell viability tests are incomplete

Major Comments:

3.2.2 XRD reults:

Laponite shows an intence diffraction peak at 2theta=3.15 due to the (001) planes. https://doi.org/10.1039/C7RA02017A

In the present results there are no peaks in the laponite graph. Alo the evidence of drug intercalation is shown in the literature from the increase of this interplanar distance taken from the XRD graphs. The authors should check again the XRD of laponite and explain in more details the drug intercalation

Is it possible to show in a histogram  “cell viability %” versus each sample?

Minor Comments:

Figure 5: Y axis : Show the endotherm direction

Figure 4: Y axis: replace weight % with “mass %”

Figyre 6: Add the Miller indices on the graph

Figure 7: Label the major peaks

Two reviews articles that the authors can  find interest information about laponite and drug delivery are:

DOI: 10.1002/adhm.202102054   and https://doi.org/10.3390/ph16060821

Author Response

Thank you kindly for your support and collaboration.

Reviewer 2 Report

pharmaceutics-2368666

Improving Riparin-A dissolution through a laponite based nanohybrid

The manuscript by Gomes et al. described the development of a nanohybrid of riparin and laponite to improve the dissolution of riparin. The nanohybrid was characterized using XRD, FTIR, TGA/DTG, DSC, and dissolution tests. Although the manuscript presents some data supporting the formation of riparin – laponite nanohybrid, this study is still in a preliminary phase. Below are some specific comments to improve this manuscript

1. What are the limitations of previous approaches to increase riparin solubility and dissolution that lead to the requirement of developing this nanohybrid? In the end, did the nanohybrid solve the problem and overcome previous approaches?

2. Factorial design: it is inappropriate to choose laponite 0%. Without laponite, changing the stirring time on riparin alone has no meaning. Also, what is “laponite concentration” referred to? The ratio of 2 components is critical, not “laponite concentration”.

3. The complex is called “nanohybrid”. Is it in nanosize? There are no data supporting this.

4. Does the stirring time too long? Will ethanol evaporate during the stirring (up to 72 h)?

5. Please revise all the decimal separators. “,” should be replaced with “.”

6. Please compare the solubility of riparin with the presence of laponite to those in previous studies. Is it considerably higher than the others?

7. Solid state characterization of the physical mixture, nanohybrid, and raw materials: in MF, the peaks of Rip-A disappear, which could be due to the suppression by laponite or the interaction between the 2 components.

8. In vitro dissolution study: what is the amount of riparin used for the dissolution study?

9. In vivo PK study should be included.

Minor editing of English language required.

Author Response

(The authors gave the same response as above.)

Round 2

Reviewer 1 Report

The revised manuscript is improved and can be accepted for publication

Author Response

(The authors gave the same response as above.)

Reviewer 2 Report

The manuscript was revised accordingly. However, there are some issues to consider as follows.

1. Explanations in the response letter should be included in the manuscript.

2. Preparation of the physical mixture and nanohybrid: the authors did not mention the amount of Rip-A as well as the ratios of Rip-A/ Lap.

3. Figure 8: please present data as means +/- SDs.

4. Figure 9: please clarify the statistical test and significant symbol.

5. Factorial design: please include the raw data (solubility of Rip-A in nine different experiments) in a Supplementary file.

Author Response

(The authors gave the same response as above.)

Round 3

Reviewer 2 Report

The manuscript was appropriately revised and can be accepted for publication.